# Hybrid Composite-Metal Stack Drilling with Different Minimum Quantity Lubrication Levels

**DOI:** 10.3390/ma12030448

**Published:** 2019-02-01

**Authors:** J. Fernández-Pérez, J. L. Cantero, J. Díaz-Álvarez, M. H. Miguélez

**Affiliations:** Department of Mechanical Engineering, Universidad Carlos III de Madrid, Avda. Universidad 30, Leganés, 28911 Madrid, Spain; jfperez@pa.uc3m.es (J.F.-P.); jodiaz@ing.uc3m.es (J.D.-Á.); mhmiguel@ing.uc3m.es (M.H.M.)

**Keywords:** hybrid stacks drilling, minimum quantity lubrication, hole quality, tool wear

## Abstract

Hybrid stack drilling is a very common operation used in the assembly of high-added-value components, which combines the use of composite materials and metallic alloys. This process entails the complexity of machining very dissimilar materials, simultaneously, on account of the interactions that are produced between them, during machining. This study analyzed the influence of Minimum Quantity Lubrication (MQL) on the performance of diamond-coated carbide tools when drilling Ti/carbon fiber reinforced plastics (CFRP)/Ti stacks. The main wear mechanism observed was diamond-coating detachment, followed by fragile breaks in the main cutting-edge. The tests done with the lower lubrication levels have shown an important adhesion of titanium (mainly on the secondary cutting-edge) and a higher friction between the tool and the workpiece, producing higher temperatures on the cutting region and a thermal softening effect on the workpiece. These phenomena affect the evolution of cutting power consumption with tool wear in the titanium layer. Regarding the quality of the test specimen, no significant differences were observed between the lubrication levels tested.

## 1. Introduction

The use of hybrid stacks for the design of high-added-value components has skyrocketed in the last decade, especially in the aeronautical industry. These stacks are composed of different materials, mainly composites and metals. The most common of these are titanium alloys (Ti) and carbon fiber reinforced plastics (CFRP) [1]. These materials stand out because of their excellent properties, and combining them provides the component with their individual characteristics. In contrast, interactions between these materials may produce a galvanic corrosion, so the use of sealants is required [2].

More than the 50% of the structural weight of some commercial aircrafts—such as the Boeing 787 and A350—is made of composite materials. Most of these are CFRP, and about 15% is made of Ti alloys [3]. To ensure the proper assembly of two components in the aerospace industry, the use of mechanical joints is still unavoidable. Drilling operations have a remarkable impact on the overall manufacturing cost, so the optimization and control of this process may have a big economic impact.

Hybrid stack drilling is a complex procedure due to the disparate nature and machinability of the materials involved. Composite fiber reinforced plastics are anisotropic, made of two faces—the fibers or reinforcements (which have a fragile behavior and low thermal conductivity) and the matrix or binder (which is more ductile). Its machining is characterized by the intermittent fracture of the fibers [4]. On the other hand, titanium alloys have a very high toughness, low elastic modulus, and resistance at high temperatures. Furthermore, they present poor machinability due to their high strength, very low thermal conductivity, and high chemical affinity with most of the cutting tools materials [5].

The study carried out by Wang et al. [6] analyzed the different wear mechanisms produced by each individual layer, when drilling hybrid stacks. They found that flank and chipping were the main mechanisms acting on Ti and edge-rounding in CFRP, which complicates the characterization of the process and the establishment of the tool life. Ramulu et al. [7] highlighted that carbides are the most appropriate materials for these processes, based on the life of the tools and the quality of the component. Park et al. [8] observed that the main wear mechanisms of uncoated carbide tools in composite/titanium stacks is flank-wear, due to the hardness of the carbon fibers and Ti adhesion. Fernandez–Vidal et al. [9] also studied the behavior of uncoated carbide tools, but on composite and aluminum stacks. They have shown that, in this case, bond-wear dominates over the abrasion of the tool surface. Rezende et al. [10] tested several cutting geometries in an aluminum/composite sandwich stack and found that the brad and spur geometry produced lower thrust forces and shorter burrs. Other authors described the influence of the tool wear in the modification of the cutting geometry and the associated machining-induced damage on the composite layers [11].

Along with the possible defects produced during the manufacturing process [12], drilling operations might cause damage on the composite layers in the form of delamination, thermal degradation, and fuzzing. When dealing with hybrid stacks, the interaction with metallic layers can further affect the quality of the hole [13]. The high temperatures produced during Ti drilling can induce thermal damage on the hole, by burning the matrix. Additionally, the metallic chips or the burrs can affect the surface quality. Furthermore, the different elastic modulus of the materials can produce variable diameters in each layer, affecting the structural integrity of the mechanical joint [14]. A completed analysis of the quality requirement in hybrid stack components was made by Shyha et al. [15].

The use of cutting fluids can help the machining process by dissipating the heat produced at the cutting region, and its use is a widespread practice [16]. The environmental impact of cutting fluids, its influence on the working environment, and its cost, justify the interest in reducing its utilization [17]. Furthermore, in hybrid composite-metal stack drilling processes, the use of conventional fluids to avoid the contamination of the composite layers is not allowed. For these operations, the Minimum Quantity Lubrication technique (MQL) is used. It is characterized by a high pressure flow of air with a small quantity of pulverized oil, which is applied directly to the cutting region, through the lubrication channels inside the tool [18], only when the tool drills through the metallic layers. To the extent possible, it is interesting to reduce the amount of oil used, to avoid cleaning operations after drilling. Several studies [19] have shown the influence of applying MQL on hole quality, in different machining processes. Giasin et al. [20] analyzed the impact of MQL lubrication in a glass fiber and aluminum stack drilling process and they observed a reduction of the burr height, at the exit of the stack, in dry conditions. Bhowmick et al. [21] studied the process of MQL drilling of a cast magnesium alloy, in dry conditions, and they found a reduction in the built-up edge formation and a reduction in the torque and thrust force. Furthermore, a softening effect of the workpiece—owing to high temperatures—was produced during dry drilling but not with MQL.

The high cost of the tools used and the number of mechanical joints required in the assembly of the high value components in the aeronautical industry makes it very difficult to complete the operations with adequate levels of competition. Furthermore, the restrictive tolerances require periodical quality controls to avoid the rejection of the component.

Hence, the objective of this study was to analyze the influence of the MQL level—through the interior of the tool—on the performance of the hybrid stack drilling process, with diamond-coated carbide drill bits. The quality of the component, the tool wear, and the evolution of the power consumption during the process were evaluated.

## 2. Materials and Methods

The experimental equipment used to perform the test and analyze the quality of the coupons is presented below.

### 2.1. Workpiece Materials and Cutting Tools

The cutting tools tested were made of a carbide substrate K10, with a diamond coating from HAM Präzision. Figure 1 shows one of the drill bits tested. These tools were specially designed following the Airbus Getafe specifications. They had two cutting edges, with a point angle of 140°, a diameter of 7.6 mm, and a split-point geometry. Two lubrication channels were located at the frontal relief surface of each edge.

The specimen tested was a hybrid stack used in the aeronautical industry, with a Ti/CFRP/Ti configuration. On one hand, the Ti alloy used on this coupon was Ti6AlV. On the other hand, the composite material was a carbon-fiber-reinforced polymer, composed of multiple unidirectional layers with different orientations. These were covered with an epoxy pre-impregnated copper foil, in the upper part, and with a prepreg made from a glass fiber fabric pre-impregnated with epoxy, in the lower part.

### 2.2. Machining Conditions

The drilling tests were done with the machines, tools, materials, and cutting conditions used for the assembly process of aeronautical components. Hence, the results presented can be directly applied to industrial production conditions.

The machine used was a computer numerical control (CNC) gantry drilling machine. The spindle head was a MFW 1406/24/2 from Fischer (Switzerland), with a nominal power of 15.200 W and a feed movement motor 1FK7 from Siemens (Berlin, Germany), with a nominal power of 940 W. The MQL lubrication system used was a Lubrix V5, and the lubricant oil used was Accu-Lube LB5000 (Maulbronn, Germany).

Two MQL lubrication levels, corresponding to the nominal conditions of the equipment, were tested. One was a low level, with an oil flow of 3 mL/h, and another was a high level, with a flow of 15 mL/h. For each condition, 40 holes were made with a new tool.

Table 1 shows the main cutting parameters used and the thickness of each layer of the stack. Different cutting conditions were used for the composite and metallic layer, due to their disparate machinability properties. For Ti drilling, a smaller cutting speed, MQL lubrication, and peck drilling were required to mitigate the formation of long and continuous chips, which can damage the composite layer and promote tool wear. On the CFRP layer, the lubrication flow had to be interrupted, since it was not needed for this material, and the excess oil could have affected the behavior of the composite materials. The possible influence of the residual lubrication of the CFRP layer is minimal, since the amount of oil used on the metallic layers is very low. The staking of the coupon was made by applying pressure with clamps.

The CNC machine used was equipped with sensors to independently measure the power consumptions of the spindle and the feed motors, which were closely related to the cutting torque and the thrust force, respectively. Being able to measure and analyze these variables provided important information on the process [22].

### 2.3. Hole Quality and Tool Wear Evaluation

Controlling the quality of the holes is mandatory for fulfilling the requirements which ensures a proper behavior of the manufacturing process and the mechanical joint. The tolerances of the different quality requirements depended on the applications of the process. In this case, the application was the assembly of the aeronautical components through mechanical joints. Hence, the tolerances required were considered to be “low clearance fit assemblies in high-load transfer applications”.
Diameter: The measurements were done with the digital bore gauge XT3 from Bowers (Bradford, England). Calibrated setting rings were used to check the repeatability and accuracy of the measurements and variations of ±2 µm were observed. To define each diameter, three measurements were done at the entrance and exit of each layer. The average value was taken.Hole surface quality: This was measured with a contact profilometer MARWIN XCR20 from Mahr (Esslingen, Germany), using a 5 µm styli, with a measurement speed of 0.5 mm/s. The roughness parameter used in this case was the average roughness parameter (Ra). Three consecutives measurements of the inner surface of the hole were done, taking the highest value obtained as the result was based on Standard DIN 4774, to ensure the fulfillment of the dimensional tolerance required in all cases. Additionally, the skewness parameter (Rsk) was also calculated. Figure 2 shows the contact profilometer equipment used to analyze the hole surface quality.Machining induced damage on the composite layer: Holes were visually inspected to look for damage in the form of delamination and scratching produced by metallic chips, fuzzing, or thermal degradation. No damages were observed in any of the holes analyzed in this study.Burr: This phenomenon is characteristic of the metallic layers and it is produced at the entrance and exit of the hole. A Mitotuyo 2046S dial gauge (Kawasaki, Japan) was used. Several measurements were done on the same hole, to determine the maximum value. Differences in the values obtained for each hole were in the order of 0.05 mm.

Tool wear was monitored and characterized using a scanning electron microscope (SEM) Philips XL-30 (Amsterdam, Netherlands), with an EDSDX4i system.

## 3. Results and Discussion

The influence of the lubrication level on the performance of the hybrid stack drilling processes was analyzed in terms of tool wear, hole quality, and cutting and feed movement power consumption.

### 3.1. Tool Wear Characterization

Figure 3 and Figure 4 show the condition of the tools after 40 holes. These were tested with a lubrication level of 3 and 15 mL/h, respectively. The main wear mechanisms produced in both tools were the loss of the diamond coating by delamination and fragile breakages of the main cutting edges. Furthermore, Ti adherence was found in both conditions, but a stronger built-up edge was produced on the secondary cutting-edge of the tool tested with the lower lubrication level.

The loss of the diamond layer affected the whole length of the main cutting-edges and the secondary cutting-edge. It was produced by the high temperatures generated during the Ti layer drilling, which induced thermal stresses on the coating from the differences between the coefficients of thermal expansion of the diamond layer and the carbide substrate. Furthermore, the fluctuations of the cutting forces, during the CFRP layer machining aggravated the tool wear mechanism [23]. Therefore, the hardness of the cutting-edge regions where the coating was lost was reduced and the substrate was exposed to the abrasiveness of the carbon fibers [24].

Regarding the fragile breaks, it can be seen in Figure 3 and Figure 4 that they were produced from the middle region of the cutting-edge to the tip of the tool. The morphology of the damage was very similar for both lubrication levels, but its extension was larger at the lower lubrication level condition. Furthermore, the tool tested with the lower lubrication level also suffered damage on the tip of the cutting-edge (Figure 3).

The description of the wear suffered on the tools was focused on the rake surface of the main cutting-edges. Figure 5 shows a detailed view of the relief surface where a loss of diamond coating by delamination can also be seen, but not any other additional types of tool wear.

Ti has a strong affinity with the substrate material of the cutting tools, especially at high temperatures. Figure 6 shows a composition analysis of the cutting-edges of the tools tested with a lubrication level of 3 mL/h (Figure 6a) and 15 mL/h (Figure 6b). The black color corresponds to carbon components, the grey tones are the titanium alloy, and the white color indicates the tungsten of the carbide substrate. The tools tested with the lower lubrication condition suffered a greater Ti-adhesion on the rake surface of the main cutting-edge and a strong built-up edge formation on the secondary cutting-edge. The higher lubrication level reduces the cutting temperatures [21] and the coefficient of friction between the tool and the chip [25].

### 3.2. Hole Quality Analysis

Controlling the dimensional quality of the holes is important in guaranteeing the suitable behavior of the mechanical joint. Furthermore, this analysis will determine whether the hole is valid for the application or a subsequent reaming operation is required.

#### 3.2.1. Diameter Analysis

Figure 7 shows the diameter measurements. The values obtained from the upper and lower Ti layers were named Ti1 and Ti2, respectively. In this case, no differences were observed among the entrance and exit side of each Ti layer, so just one value was represented for each of them. On the other hand, in the composite layer, a different behavior was observed among the entrance and exit side of the layer, so one value for each side was represented—CFRP in and CFRP out, respectively.

The diameter of the holes in the Ti layers showed a very similar behavior, in both lubrication levels. For the lower lubrication level, a higher dispersion of the diameters of the holes on the CFRP layer was observed at the beginning of the tool’s life. This could be explained by the higher temperatures produced on the Ti layers [13], which generated longer and continuous chips. This chip morphology was not desirable since it over-sized the holes of the CFRP layer, during the chip evacuation. Once the number of holes had increased, the tool wear modified the cutting geometry towards a more negative one, producing shorter and discontinuous chips [15]. Beyond the 30th hole, the diameter of the holes obtained on the CFRP and the Ti layers became similar.

For higher lubrication levels, the diameters of the holes on the CFRP layer were similar to those in the Ti layer, and they remained nearly constant, throughout the tested lifespan of the tool.

The exit side of the hole of the CFRP layer had a greater diameter, compared to the entrance side, due to the larger influence of the chips produced in the lower Ti layer drilling, during its evacuation [26].

#### 3.2.2. Burr Height Analysis

Burr height values, represented in Figure 8, showed no variation as the tested lubrication levels changed. With the fresh tool, the height values for both tools were around 0.05 mm at the entrance of the stack, and 0.1 mm at the exit. As the tool wear evolved, the burr height at the entrance increased up to 0.15 mm and, at the exit side of the stack, it reached values close to 0.4 mm. In both cases, they were acceptable values [27] although, for assembly purposes, a deburring process would be required [28].

#### 3.2.3. Hole Surface Quality Analysis

The study on roughness shows important differences in the machining behavior between the composite material and the metal, which can be seen in Figure 9.

The chip formation mode on composite materials, which depends on the relative orientation of the cutting-edge with respect to the fibers, has a big influence on surface quality [29]. Owing to this phenomenon, the roughness obtained during the drilling processes of these materials is variable. Furthermore, interactions with the metallic chips can further deteriorate the obtained roughness [30].

More stable values of the surface quality of the Ti layers were obtained. Roughness values (Ra) of around 1 µm were found from the fresh tool, up to the 40th hole. No influence from the lubrication level or the tool wear, was observed.

The skewness analysis also showed differences between the hole surface quality of the metallic layers and the composite layers. In Figure 10, it can be seen that the parameter Rsk was always negative, for the CFRP layer, while it was around zero in the Ti layers, with negative and positive values.

### 3.3. Power Consumption Analysis

The spindle and feed motor power consumptions were recorded, individually, and these measurements were expressed as a percentage of their nominal power consumptions. Along with the net power associated to the drilling process, the signal included friction, noise, or perturbations. Thus, the values obtained were analyzed qualitatively.

Figure 11 shows the spindle power consumption raw signal obtained during the Ti/CFRP/Ti stack drilling process of the 40th hole. Noticeable fluctuations could be seen while the tool was spinning in the air, before the tool penetrated the coupon (first 10 s of the signal), which needed to be filtered out. For this reason, a low-pass filter was applied to the signal; this is represented in Figure 11 as a solid black line.

During the Ti drilling, oscillations on the power consumption signal were produced by the peck drilling cycle. The change of the cutting parameters between the Ti and CFRP layers could be observed as a sharp increase in consumption, in order to adapt the spindle speed. For the CFRP layer, the consumption was constant, since no peck drilling was done. However, there were big oscillations in the signal, owing to the fluctuation of the cutting forces produced during the fiber-reinforced composite machining [31]. After the CFRP layer, a new sharp increase of the power consumption was produced, again, to match the Ti cutting parameters.

The Ti power consumption was approximately 10% of the nominal power of the spindle motor, with very similar values on both layers, Ti1 and Ti2. Thus, hereafter, the average value of both layers was considered as the characteristic value for the Ti layers’ consumptions. At the CFRP layer, the power consumption was an order of magnitude smaller (by 1%) than the nominal power consumption.

Figure 12 shows the feed motor power consumption obtained during a Ti/CFRP/Ti stack drilling process. It had a similar trend to the spindle power consumption. However, in each peck cycle, a sharp increase of the power was produced.

It can be seen in Figure 11b and Figure 12b that the shape of power consumption in each individual peck drilling cycle followed the same behavior as that of a conventional continuous drilling process [30]. In all cases, the amplitude of the signal measured, while the tool was approaching the material, was eliminated.

The influence of the lubrication level on the evolution of the spindle power consumption with the tool wear can be seen in Figure 13a, for the Ti layers, and Figure 13b, for the composite layer. With the fresh tool, the power consumption for the Ti layer with the higher lubrication level (blue asterisks) was 8% smaller, due to the reduction of the friction coefficient between the tool and the workpiece [32].

However, this behavior was not maintained with the evolution of the tool wear. The gradient of the relationship between the spindle power consumption and the number of holes was steeper for the higher lubrication level (blue asterisks) and flatter for the lower lubrication level (red triangles). This could be related to the different Ti adhesions observed on the rake surface of the tools. At lower lubrication levels, the higher coefficients of friction between the tool and the workpiece, combined with larger temperatures, produced a modification of the chip formation mode and led to a greater influence of the thermal softening effect [33]. Therefore, the gradient of the spindle power consumption with the number of holes was less steep.

Hence, the spindle power consumption increment on the Ti layer, after 40 holes, with respect to the fresh tool, was 38% for the lower lubrication level and 76% for the higher lubrication level.

On the CFRP layer drilling process, no lubrication was used and the thermal softening effect had less influence. Thus, the increment of the spindle power consumption with the number of holes was higher for the lower lubrication level, owing to the higher tool wear suffered from the metallic layers (fragile breakages). Furthermore, no differences were observed with the fresh tool, which indicated that the residual oil did not have an influence on the CFRP layer drilling. In this case, the increment of the spindle power consumption, after 40 holes, with respect to the fresh tool, was 88% for the lower lubrication level and a 56% for the higher lubrication level.

Feed movement power consumption can be seen in Figure 14. The behavior was very similar to the cutting movement power consumption. With the fresh tool, the power of the higher lubrication level was 6% smaller due to the reduction of the friction coefficient. This tendency was reversed, and the evolution of the feed movement power consumption with the number of holes was faster for the higher lubrication level (blue asterisks) and more gradual for the lower lubrication level (red triangles). This was also explained by the phenomenon related to the modification of the chip formation mode, and the thermal softening effect. In this case, after 40 holes, the feed movement power consumption increased by 24% and 61% for the 3 mL/h and the 15 mL/h lubrication levels, respectively.

Regarding the feed movement power consumption on the CFRP layer, no analysis was carried out since the magnitude of the signal was too small. There was not enough sensibility in order to draw solid conclusions.

## 4. Conclusions

The use of the MQL lubrication system during a Ti/CFRP/Ti hybrid stack drilling process was analyzed. With a minimum amount of lubrication, the behavior of the process during the metallic layers drilling was significantly affected without having an influence on the performance of the composite layer drilling.

It was found that the main wear mechanisms suffered by the diamond-coated carbide drill bits, during this process, were the loss of the coating by delamination and fragile breaks on the middle region of the main cutting-edge. Additionally, the level of lubrication used affected the Ti adhesion to the cutting tool. For the lower lubrication level, a strong Ti adhesion was found on the rake surface of the main cutting-edge and the secondary cutting-edge, due to the higher friction coefficient and temperatures produced. In contrast, this was not produced for the higher MQL level.

The differences induced by the level of lubrication also affected the evolution of the cutting and feed movement power consumption on the Ti layers. With the fresh tool, less consumption was found for the lower lubrication level but, as the tool wear increased, this tendency was reversed due to the differences in the Ti adhesion suffered by the tools and the influence of the thermal softening effect. In contrast, the level of lubrication did not directly affect the drilling process during the CFRP layer. However, in this material, the gradient of the relationship between the spindle power consumption and the number of holes was steeper for the lower lubrication level, due to the higher wear suffered by the Ti layers.

No significant differences were observed on the quality of the components between the lubrication levels analyzed. However, a higher dispersion of the diameter of the holes in the composite layer for the lower lubrication condition was produced.

## Figures and Tables

**Figure 1 materials-12-00448-f001:**
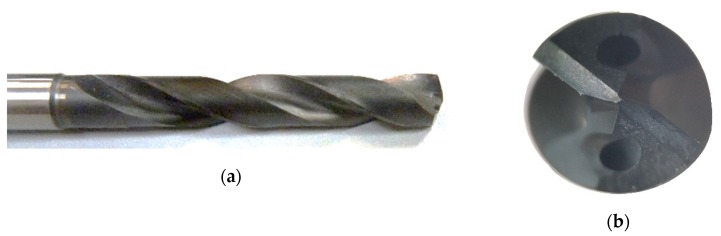
The diamond-coated carbide drill bit with a 7.6 mm diameter that were tested. (**a**) Lateral view. (**b**) Frontal view.

**Figure 2 materials-12-00448-f002:**
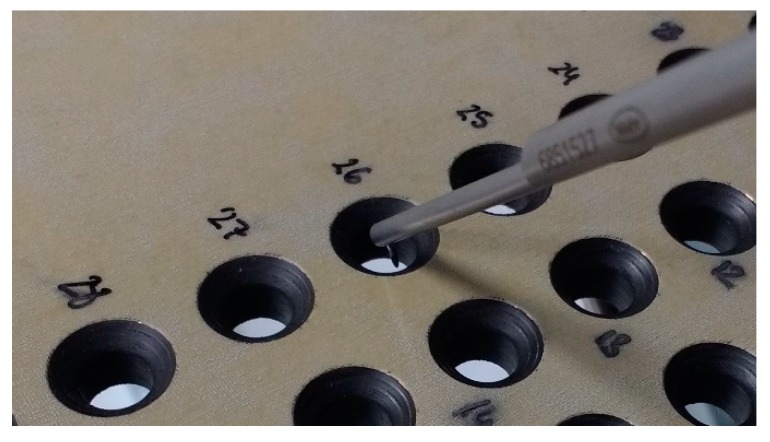
Hole profile and roughness analysis of a carbon fiber reinforced plastics (CFRP) coupon with a contact profilometer.

**Figure 3 materials-12-00448-f003:**
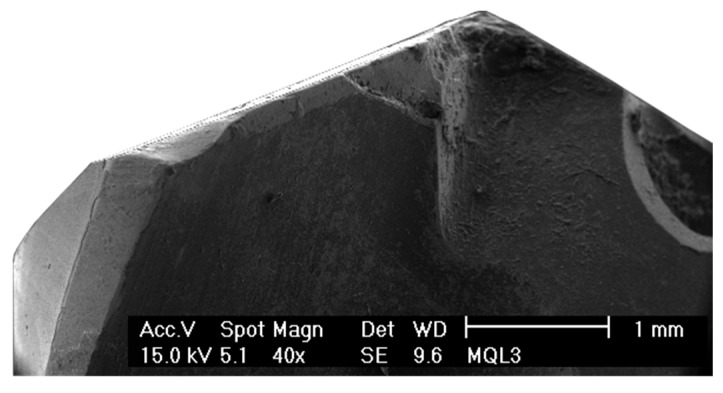
Wear of the tool tested on a Ti/CFRP/Ti stack with the lower lubrication level (3 mL/h), after 40 holes.

**Figure 4 materials-12-00448-f004:**
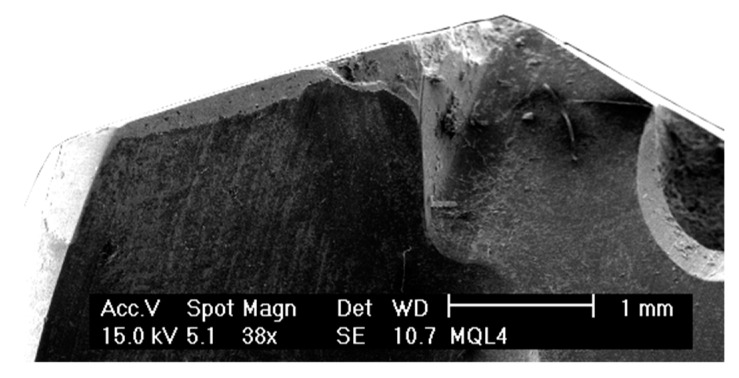
Wear of the tool tested on a Ti/CFRP/Ti stack with the higher lubrication level (15 mL/h), after 40 holes.

**Figure 5 materials-12-00448-f005:**
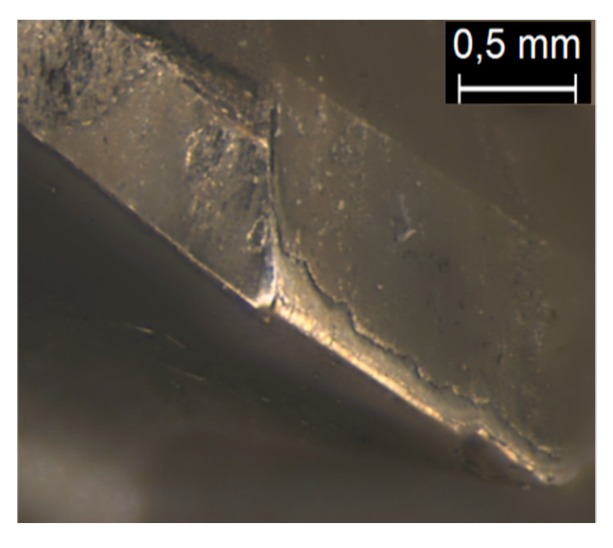
Detailed view of the relief surface of the tool with the lower lubrication level, after 40 holes.

**Figure 6 materials-12-00448-f006:**
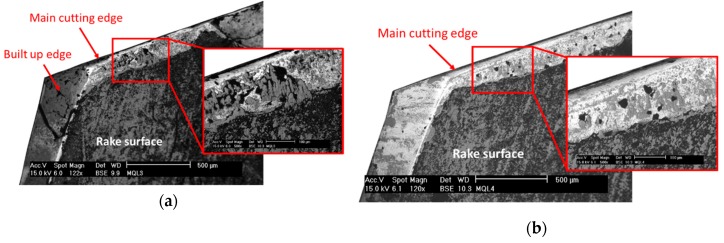
Composition analysis of the rake surface, after 40 holes, in a Ti/CFRP/Ti coupon. The black color corresponds to the carbon components, the grey tones are titanium alloy, and the white indicates the tungsten of the carbide substrate. (**a**) A 3 mL/h lubrication level. (**b**) A 15 mL/h lubrication level.

**Figure 7 materials-12-00448-f007:**
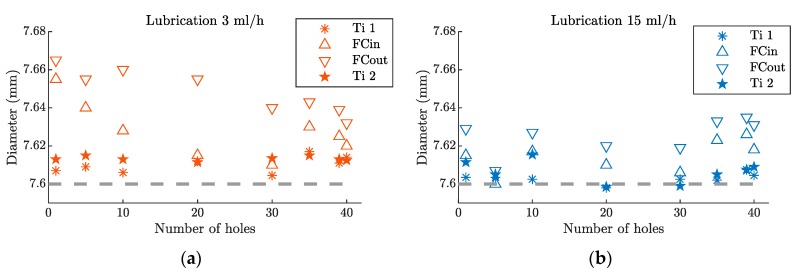
Hole diameter evolution, after 40 holes, in a hybrid Ti/CFRP/Ti stack. The dashed line represents the nominal diameter of the tool with (**a**) a lubrication level of 3 mL/h and (**b**) a lubrication level of 15 mL/h.

**Figure 8 materials-12-00448-f008:**
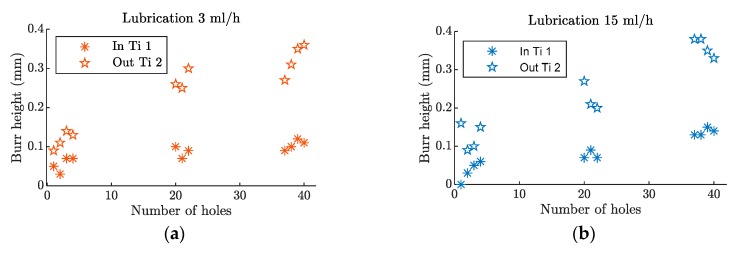
Burr height evolution, after 40 holes, in a hybrid Ti/CFRP/Ti stack with a (**a**) lubrication level of 3 mL/h and a (**b**) lubrication level of 15 mL/h.

**Figure 9 materials-12-00448-f009:**
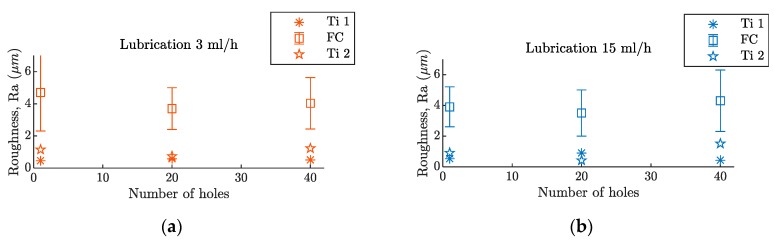
Roughness evolution, after 40 holes, in a hybrid Ti/CFRP/Ti stack with a (**a**) lubrication level of 3 mL/h and a (**b**) lubrication level of 15 mL/h.

**Figure 10 materials-12-00448-f010:**
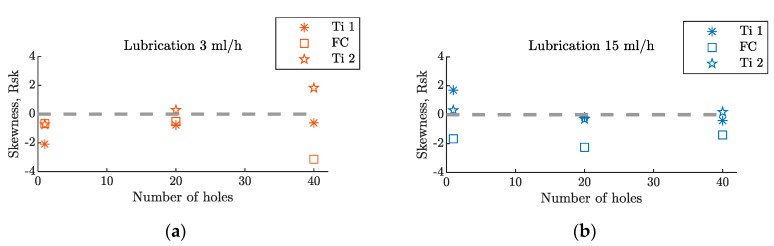
Skewness evolution, after 40 holes, in a hybrid Ti/CFRP/Ti stack with a (**a**) lubrication level of 3 mL/h and a (**b**) lubrication level of 15 mL/h.

**Figure 11 materials-12-00448-f011:**
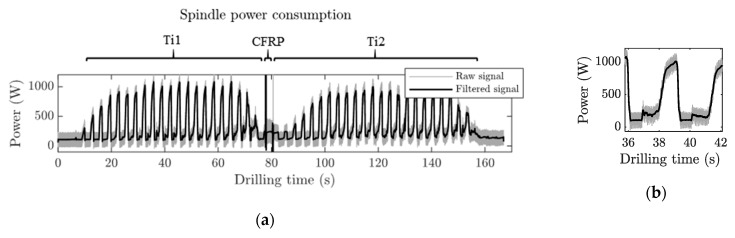
Spindle power consumption signal of hole 40 with the higher lubrication level (15 mL/h) recorded during Ti/CFRP/Ti stack drilling. (**a**) The complete signal of the drilling process. (**b**) A detailed view of a peck drilling cycle on the first Ti layer.

**Figure 12 materials-12-00448-f012:**
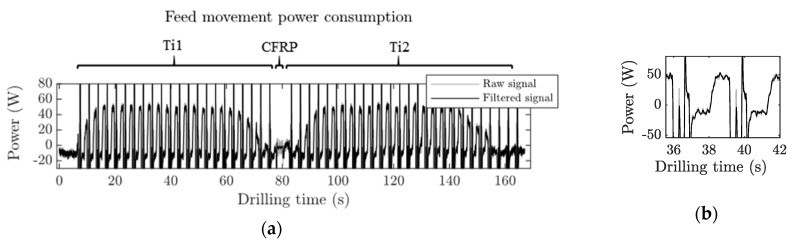
Feed movement power consumption signal of the 40th hole with the higher lubrication level (15 mL/h) recorded during the Ti/CFRP/Ti drilling. (**a**) The complete signal of the drilling process. (**b**) A detailed view of a peck drilling cycle on the first Ti layer.

**Figure 13 materials-12-00448-f013:**
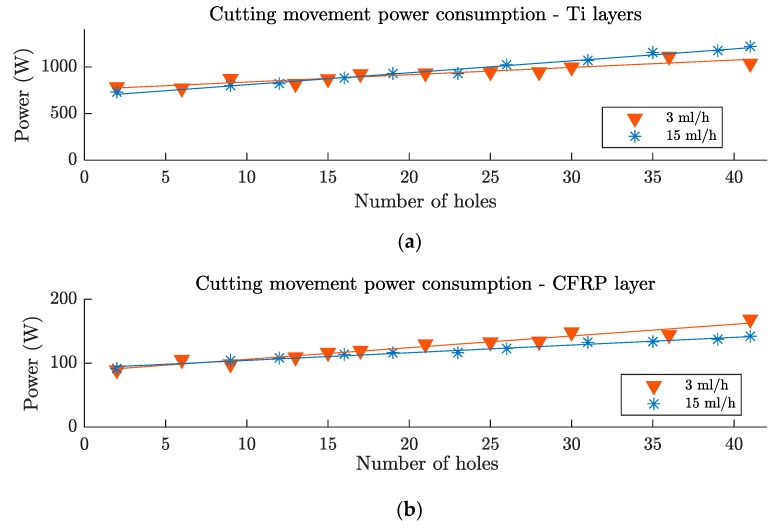
Spindle power consumption, after 40 holes, for a Ti/CFRP/Ti stack on (**a**) the Ti layers and (**b**) on the CFRP layer.

**Figure 14 materials-12-00448-f014:**
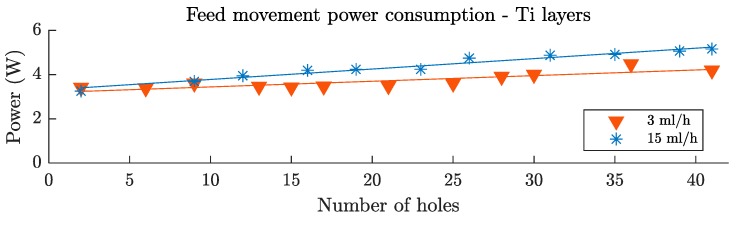
Feed movement power consumption on the Ti layer, after 40 holes, for a Ti/CFRP/Ti stack drilling process.

**Table 1 materials-12-00448-t001:** The stack configuration and cutting parameters tested.

Material	Thickness (mm)	Cutting Speed (m/min)	Feed (mm/rev)	Minimum Quantity Lubrication (MQL) Lubrication
Ti	9.3	15	0.05	Yes
CFRP	8.4	70	0.05	No
Ti	9.3	15	0.05	Yes

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
