# Peer review of "Hybrid Composite-Metal Stack Drilling with Different Minimum Quantity Lubrication Levels"

_materials, 2019, doi:10.3390/ma12030448_

Round 1

Reviewer 1 Report

*Overall a great paper that goes in depth and explains throughly and clearly the methods used and the results reached. I would suggest studying the following publications as a way to enrich the mentioned research topics covered in the submitted paper:

·        K. Anyfantis, P. Stavropoulos, G. Chryssolouris, "Fracture mechanics based assessment of manufacturing defects laying at the edge of CFRP-metal bondlines", Production Engineering, Vol. 12, No. 2, pg. 173-183, (2018)

·        P. Stavropoulos, A. Papacharalampopoulos, E. Vasiliadis, G. Chryssolouris, "Tool wear predictability estimation in milling based on multi-sensorial data", The International Journal of Advanced Manufacturing Technology, Volume 82, Issue 1, pp. 509-521,  (2016)

 Additionally, a few english corrections to an otherwise properly presented piece of reseach work.

Author Response

Thank you for the dedication to review the paper. All comments were taken into account to improve the quality of the research and they will be taken into account for future works. The document has been updated with your comments and the modifications introduced have been highlighted with the application “Track Changes” in Microsoft Word.

Hereafter, each comment is answered individually.  

Comments and Suggestions for Authors from Reviewer 1:

Overall a great paper that goes in depth and explains throughly and clearly the methods used and the results reached.

-          I would suggest studying the following publications as a way to enrich the mentioned research topics covered in the submitted paper:

o   K. Anyfantis, P. Stavropoulos, G. Chryssolouris, "Fracture mechanics based assessment of manufacturing defects laying at the edge of CFRP-metal bondlines", Production Engineering, Vol. 12, No. 2, pg. 173-183, (2018)

o   P. Stavropoulos, A. Papacharalampopoulos, E. Vasiliadis, G. Chryssolouris, "Tool wear predictability estimation in milling based on multi-sensorial data", The International Journal of Advanced Manufacturing Technology, Volume 82, Issue 1, pp. 509-521,  (2016)

Response: The references mentioned have been revised and included in the document to enrich the research discussed.

-          Additionally, a few english corrections to an otherwise properly presented piece of reseach work

Response: A professional English editing service has been hired in order to improve the general quality of the document.  

Reviewer 2 Report

Article is very interesting and written very well.

Comments and Suggestions for Authors (the proposed amendments):

- not shown drills geometry (Fig. 1),

- only the Ra parameter was measured and presented, it is possible to assess the performance properties of hole surface by providing for example, parameters Rsk, Rku and 3D parameters (Sa, Sz),

- the article can be extended by analysis of eg the components of the cutting force and the cutting torque, vibrations (accelerations and displacements) which are important indicators describing the
machinability of materials,

- graphical experimental set-up need to show in the article.

Author Response

Thank you for the dedication to review the paper. All comments were taken into account to improve the quality of the research and they will be taken into account for future works. The document has been updated with your comments and the modifications introduced have been highlighted with the application “Track Changes” in Microsoft Word.

Hereafter, each comment is answered individually.

Comments and Suggestions for Authors from Reviewer 2:

Article is very interesting and written very well.

-          Not shown drills geometry (Fig. 1),

Response: A new image has been added as Figure 1. (b). It shows the frontal view of the cutting tool where the cutting geometry can be observed

-          Only the Ra parameter was measured and presented, it is possible to assess the performance properties of hole surface by providing for example, parameters Rsk, Rku and 3D parameters (Sa, Sz),

Response: The analysis of the hole surface quality was improved with the discussion of the skewness parameter Rsk. A new Figure with the evolution of the Rsk parameters was also included. Additionally, it would be very interesting to include an analysis of 3D parameters. However, they could not be obtained since the required equipment was not available.

-          The article can be extended by analysis of eg the components of the cutting force and the cutting torque, vibrations (accelerations and displacements) which are important indicators describing the machinability of materials,

Response: Indeed, the magnitudes mentioned have great interest and provides valuable information of the cutting process. However, it is not possible to include these magnitudes in the article because the tests were performed on industrial machines which do not have this type of monitoring equipment. Nevertheless, the use of industrial equipment for testing allows to simulate reliably the conditions found in a real production environment and facilitates the technological transfer between researchers and industry.

-          Graphical experimental set-up need to show in the article.

Response: It has been included a Figure of the profilometer equipment which was used to measure the roughness of the coupons. Unfortunately, due to the confidentiality rules established in the facilities of the company Airbus, it is not possible to show additional images of experimental equipment used in drilling tests.

Reviewer 3 Report

This very interesting paper deals with the influence of different minimum quantity lubrication during drilling hybrid composite- metal stacks. Tool wear, hole deflections, burr heights, roughness and power consumption were analyzed and give complete results in order to discuss of what happened during drilling this composite material.

From these previous comments, this manuscript has enough quality to be accepted in this form. I suggest to accept this manuscript after minor revisions as follows:

1) In section 3.3, when presenting Fig 9, Authors should specify that reader is looking at the results at 40th holes

2) Captions in Fig 9, 10, 11 and 12 are too long. All details are shown in the legends. If Authors want to specify these details, they should write it in the text.

Author Response

Thank you for the dedication to review the paper. All comments were taken into account to improve the quality of the research and they will be taken into account for future works. The document has been updated with your comments and the modifications introduced have been highlighted with the application “Track Changes” in Microsoft Word.

Hereafter, each comment is answered individually.

Comments and Suggestions for Authors from Reviewer 3:

This very interesting paper deals with the influence of different minimum quantity lubrication during drilling hybrid composite- metal stacks. Tool wear, hole deflections, burr heights, roughness and power consumption were analyzed and give complete results in order to discuss of what happened during drilling this composite material.

From these previous comments, this manuscript has enough quality to be accepted in this form. I suggest to accept this manuscript after minor revisions as follows: 

1)       In section 3.3, when presenting Fig 9, Authors should specify that reader is looking at the results at 40th holes.

Response: It has been included a comment in the line 258 to specify that the consumption signal shown in Figure 9 was recorded during hole 40th.

2)      Captions in Fig 9, 10, 11 and 12 are too long. All details are shown in the legends. If Authors want to specify these details, they should write it in the text.

Response: Redundant explanations with the legends on the Figure captions mentioned have been eliminated. Some comments have been added to the text which refers to the Figures in order to facilitate the comprehension of the document.
